# The Secretome of Parental and Bone Metastatic Breast Cancer Elicits Distinct Effects in Human Osteoclast Activity after Activation of β2 Adrenergic Signaling

**DOI:** 10.3390/biom13040622

**Published:** 2023-03-30

**Authors:** Francisco Conceição, Daniela M. Sousa, Sofia Tojal, Catarina Lourenço, Carina Carvalho-Maia, Helena Estevão-Pereira, João Lobo, Marina Couto, Mette M. Rosenkilde, Carmen Jerónimo, Meriem Lamghari

**Affiliations:** 1I3S—Instituto de Investigação e Inovação em Saúde, Universidade do Porto, 4200-135 Porto, Portugal; 2INEB—Instituto Nacional de Engenharia Biomédica, Universidade do Porto, 4200-135 Porto, Portugal; 3ICBAS—Instituto de Ciências Biomédicas Abel Salazar, Universidade do Porto, 4050-313 Porto, Portugal; 4Cancer Biology and Epigenetics Group, IPO Porto Research Center (CI-IPOP), Portuguese Oncology Institute of Porto (IPO Porto), 4200-072 Porto, Portugal; 5Department of Pathology, Portuguese Oncology Institute of Porto (IPO Porto)/Porto Comprehensive Cancer Centre (Porto.CCC), 4200-072 Porto, Portugal; 6Department of Biomedical Sciences, University of Copenhagen, Blegdamsvej 3B, DK-2200 Copenhagen, Denmark

**Keywords:** breast cancer, beta-adrenergic, sympathetic nervous system, osteoclast, proteomic

## Abstract

The sympathetic nervous system (SNS), particularly through the β2 adrenergic receptor (β2-AR), has been linked with breast cancer (BC) and the development of metastatic BC, specifically in the bone. Nevertheless, the potential clinical benefits of exploiting β2-AR antagonists as a treatment for BC and bone loss-associated symptoms remain controversial. In this work, we show that, when compared to control individuals, the epinephrine levels in a cohort of BC patients are augmented in both earlier and late stages of the disease. Furthermore, through a combination of proteomic profiling and functional in vitro studies with human osteoclasts and osteoblasts, we demonstrate that paracrine signaling from parental BC under β2-AR activation causes a robust decrease in human osteoclast differentiation and resorption activity, which is rescued in the presence of human osteoblasts. Conversely, metastatic bone tropic BC does not display this anti-osteoclastogenic effect. In conclusion, the observed changes in the proteomic profile of BC cells under β-AR activation that take place after metastatic dissemination, together with clinical data on epinephrine levels in BC patients, provided new insights on the sympathetic control of breast cancer and its implications on osteoclastic bone resorption.

## 1. Introduction

Breast cancer (BC) is the most common type of cancer diagnosed in women worldwide [1]. It is a heterogeneous disease, characterized by the primary tumor status of the estrogen receptor (ER), progesterone receptor (PR) and epidermal growth factor receptor 2 (HER2). Treatment options have improved early stage BC patient survival [2], but the metastatic spread of BC cells from the primary tumor to the distant organs dramatically reduces patient life expectancy. Notably, the hormone receptor status of the primary tumor correlates with specific tropism towards different organs [3]. Bone is the most common site of metastasis, with around 70% of late-stage BC patients presenting bone metastatic foci that lead to skeletal complications, such as increased fracture risk, hypercalcemia and severe bone pain [4].

Bone homeostasis is maintained through the coupling of bone matrix deposition by osteoblasts, the cells of mesenchymal origin, and bone resorption by osteoclasts, specialized multinucleated cells able to degrade the old or damaged mineralized matrix [5]. Before and during the colonization of bone, BC cells sequester the normal bone remodeling process and shift it towards increased bone resorption [6]. Osteoclast overactivation, on the other hand, leads to the release of embedded bone matrix proteins, such as TGF-β and insulin-like growth factor (IGF), that fuel the growth of BC cells and perpetuates a “metastatic vicious cycle” of bone destruction [7].

Other players might have an important role in BC bone metastasis. Sympathetic nervous system (SNS) activity is often exacerbated in several types of cancer [8,9], and multiple pharmacological in vitro studies with human cells and murine ex vivo bone cultures have implicated the SNS in extravasation and colonization of the bone in prostate cancer [10,11]. The SNS is involved in the “fight or flight” response to stress and its main effector neurotransmitters are the catecholamines norepinephrine and epinephrine, which are released in sympathetic nerve terminals in target peripheral tissues or in the circulation by the adrenal gland, respectively [12]. Epinephrine and norepinephrine bind to α and β adrenergic receptors (α/β-AR), a family of nine receptors commonly expressed in bone cells and BC cells [13]. In particular, the seminal work by Campbell and colleagues has associated β_2_-AR signaling to increased BC bone colonization and degradation in vivo [14]. In vivo xenograft denervation studies have also highlighted the role of β_2_-AR in BC metastatic spread and immune regulation [15]. Both reports are in agreement with previous epidemiologic studies that have emphasized β_2_-AR targeting drugs as potentially beneficial in BC patient survival and recurrence [16,17,18]. Nevertheless, other retrospective studies have found no significant correlation between the use of β_2_-AR targeting drugs and improved BC survival [19,20,21]. High β_2_-AR expression in tumors of BC patients was recently associated with longer disease-free survival [22], and thus the effect of SNS activity in BC survival and recurrence remains controversial. An improved understanding of the SNS control of the role of BC in the modulation of the bone metastatic niche, in particular through the β_2_-AR signaling, are urgently needed.

In this study, we focused on the effect of β_2_-AR signaling on the crosstalk between BC cells and human osteoclasts/osteoblasts, mimicking interactions occurring at earlier stages of BC and during metastatic colonization. In addition, we measured the epinephrine levels in the plasma of primary and advanced BC patients. Our findings complement the current knowledge on the dynamics of adrenergic control of the metastatic bone niche, highlighting the changes in proteomic profile that take place during metastatic trait acquisition in BC.

## 2. Materials and Methods

### 2.1. Materials

Isoproterenol hydrochloride (ISO, Cat#I6504) and ICI 118,551 (ICI, Cat#I127) were purchased from Sigma-Aldrich (Lyon, France). All cell culture media, penicillin/streptomycin solution (pen/strep, Cat#15140122) and trypsin (0.25% *w*/*v* trypsin, 0.1% *w*/*v* glucose and 0.05% EDTA in PBS, Cat#25200056) were purchased from Gibco, Thermo Fisher Scientific (Paisley, Scotland). FBS was purchased from Biowest (Cat#S181B-500, Nuaillé, France). Recombinant human macrophage colony stimulating factor (rhM-CSF) and recombinant human RANKL (rhRANKL) were purchased from R&D Systems (Cat#216-MC-025 and Cat#390-TN-010, respectively, Milan, Italy). Epinephrine levels in the plasma of BC patients were quantified by ELISA (Abnova, Cat#157KA1877, Taoyuan, Taiwan), according to the manufacturer’s instructions.

### 2.2. Human Patients

Peripheral blood samples from control individuals (*n* = 12), stage IA BC patients (*n* = 12) and stage IV BC patients (*n* = 12) were collected at the Instituto Português de Oncologia (IPO) Porto. Briefly, peripheral blood was collected into EDTA-containing tubes and centrifuged at 2000 rpm for 10 min at 4 °C. Plasma was immediately separated, aliquoted into 1.5 mL tubes and properly stored at −80 °C until further use. All the procedures were conducted following approval by the institutional ethics committee (Comissão de Ética para a Saúde, CES-IPOFG-EPE 019/08 and CES 120/015) and sample collection was performed in accordance with the Declaration of Helsinki. Informed consent was obtained from all individual participants included in the study.

All stage IV BC patients presented distant site metastasis at diagnosis, with the bone as the most common site. A detailed description of the cohort (diagnosed between October 2015 and May 2018) is available in Appendix A.

### 2.3. Cell Culture

#### Parental and Bone Metastatic BC Cell Lines

The human breast carcinoma cell line MDA-MB-231 (MDA-231, henceforth RRID:CVCL_VR35) and its bone tropic variant, the MDA-MB-231-BoM 1833 cell line (MDA-1833, henceforth RRID:CVCL_DP48) were obtained from Dr. Joan Massagué (Memorial Sloan-Kettering Cancer Center, New York, NY, USA). Cells were expanded in standard T75 flasks (Sarsted) in DMEM High Glucose with 10% FBS and 1% Pen/Strep at 37 °C and 5% CO_2_ in a humidified incubator, changing the medium twice a week until they reached 80% confluence.

#### Human Primary Osteoblasts

Human osteoblasts were obtained from patients undergoing hip or knee replacement surgeries (following approval by the ethical committee in Hospital São João, protocol number 143/19), as previously described [23]. Informed consent was obtained from all donors before collection. Briefly, the trabecular bone was washed in PBS and cut into small pieces, which were then vigorously shaken with PBS to completely remove the remaining bone marrow. Seven to eight bone pieces were then placed in each well of a six-well plate and fixed in place with a metal grid to prevent movement during culture. Bone pieces were cultured at 37 °C and 5% CO_2_ in a humidified incubator for 14 days in DMEM Low Glucose (Cat#21885-108) supplemented with 10% FBS, 50 µg/mL ascorbic acid, 10 mM β-glycerophosphate and 10 nM dexamethasone (Sigma-Aldrich, Cat#49752, Cat#G-9891, Cat#D1756, respectively), changing the medium at day 7. Cells began to migrate from the bone to the culture plate after 7 days, and at 14 days of culture the metal grid and bone pieces were removed. The outgrowth cells were expanded until confluence was reached, changing the medium twice a week.

#### Human Primary Osteoclasts

Human CD14^+^ monocytes were isolated from buffy coats of healthy female blood donors, as previously described [24]. Briefly, peripheral blood mononuclear cells (PBMCs) were separated using gradient centrifugation in Ficoll-Paque Plus (GE Healthcare, Cat#17-1440-03, Lyon, France). PBMCs were then resuspended in 0.5% Biotin-free BSA (Sigma-Aldrich, Cat#A4919, Lyon, France) and 2 mM EDTA in PBS incubated in BD IMagTM anti-human CD14 magnetic particles (BD-Biosciences, Cat#557769, Mountain View, CA, USA) and magnetically separated according to the manufacturer’s instructions. CD14^+^ cells were seeded in T75 flasks in α-MEM (Cat#41061-029) supplemented with 10% FBS, 1% pen/strep (Osteoclast Medium) and 25 ng/mL of rhM-CSF at 5% CO_2_ at 37 °C in a humidified incubator. After 2 days, the media were changed to Osteoclast Medium supplemented with 25 ng/mL rhM-CSF and 25 ng/mL rhRANKL and cells were cultured for up to 7 days, changing media twice, until cells were morphologically differentiated.

### 2.4. Functional Studies

#### Activation of β_2_-AR Signaling in BC

MDA-231 and MDA-1833 cells were seeded at a density of 10^4^ cells/cm^2^ and left to proliferate for 24 h. Cells were then serum-starved and stimulated with the β-AR agonist ISO (1 μM) or with PBS as a vehicle control. To block β_2_-AR activation, the selective β_2_-AR antagonist ICI (1 μM) was added 45 min prior and also simultaneously as ISO to ensure the complete blockage of the signaling (Appendix A). These concentrations promoted cAMP accumulation in β_2_-AR-expressing cancer cells, and ICI addition completely blocked β_2_-AR signal transduction (Appendix A). Cells were incubated for 24 h, after which the conditioned media was collected, centrifuged at 4 °C at 400× *g* to pellet cell debris and frozen at −80 °C until further use.

#### Osteoclast Differentiation following Exposure to BC Cell Secretome

At day 5 of culturing, pre-osteoclasts were detached with Accutase (Gibco, Cat#A11105-01, Paisley, Scotland) for 10 min at 37 °C, centrifuged and seeded in 96-well tissue culture polystyrene (TCPS) (Orange Scientific, Braine-l’Alleud, Belgium) plates at a density of 20,000 cells/well in Osteoclast Medium with 25 ng/mL rhM-CSF and rhRANKL. Cells were left to adhere for 4 h and were stimulated with BC cell secretome at a 1:1 ratio (Osteoclast Medium:BC secretome), supplemented with 25 ng/mL rhM-CSF and rhRANKL. Osteoclasts were left to differentiate until day 9. Medium/Secretome was changed on day 7 of culturing.

After exposure to BC secretome, osteoclast differentiation was assessed. Cells were washed with PBS and fixed in 4% paraformaldehyde (PFA, Cat#158127, Lyon, France) for 10 min at room temperature (RT). Following another washing step with PBS, cells were permeabilized with cold 1% (*v*/*v*) Triton X-100 (Sigma-Aldrich, Cat#X100)) in PBS for 5 min. Unspecific binding was then blocked with 0.1% BSA (Sigma-Aldrich, Cat#05482, Lyon, France) solution in PBS for 1 h at 37 °C. Samples were incubated with HCS CellMask™ Deep Red Stain (Invitrogen, 1:20,000 dilution, Cat#H32721) and counterstained with DAPI (Sigma-Aldrich, 1:1000 dilution, Cat#D9542, Lyon, France) for 1 h at RT protected from light. Samples were washed twice with PBS to remove excess staining and kept at 4 °C until imaging. Images were acquired using an IN Cell Analyzer 2000 (GE Healthcare, Issaquah, WA, USA) high content screening imaging system. Six images per well were randomly acquired, on the DAPI and Cy5 channel, with a Nikon 10X/0.45 Plan Apo objective. After image acquisition, a Pixel Classification workflow of the Ilastik toolkit (v. 1.3.3 , RRID:SCR_015246) [25] was used for osteoclast cytoplasm and nuclei segmentation with σ values of 0.7, 1.0, 1.6 and 3.5 for the Gaussian Smoothing feature and 0.7, 1.0 and 1.6 for the remaining features: the Laplacian of Gaussian, Gaussian gradient magnitude, difference of Gaussian, structure tensor eigenvalues and Hessian of Gaussian eigenvalues. Up to 15 images were used in the pixel classification training, and batch processing of the images generated probability maps used later for segmentation. Next, both raw nuclei and cytoplasm images as well as Ilastik generated probability maps were uploaded into CellProfiler™ (v. 4.0.6., RRID:SCR_007358) [26]. A workflow was built in order to identify the nucleus and cytoplasm of all cells. Briefly, nuclei were segmented by manual thresholding applied to the previously generated probability masks. Then, the cell cytoplasm was segmented with the same method through a propagation algorithm of the cytoplasm mask starting from the segmented nuclei until the edges of the cytoplasm map. Cell cytoplasm and correspondent nuclei were matched, and the analysis pipeline ended with an automatic reporting of the number of cells and nuclei per cell were detected in each image (Appendix A). The number of osteoclasts with more than three nuclei was then quantified for each condition.

#### Osteoclast Resorption following Exposure to BC Cell Secretome

The effect of BC cell secretome on osteoclast resorption was performed in osteoclast monocultures or osteoclast/osteoblast co-culture experiments. After 9 days of culture, mature osteoclasts were detached with accutase for 10 min at 37 °C, centrifuged and seeded on top of 0.4 mm thick bone slices (boneslices.com, Jelling, Denmark) at a density of 50,000 cells per well in Osteoclast Medium with 0.5% FBS supplemented with 25 ng/mL rhM-CSF and were left to adhere for 4 h. No exogenous RANKL was added in resorption assays.

For osteoblast co-culture experiments, osteoblasts were detached with accutase for 10 min, centrifuged and seeded on top of the bone slices with the seeded osteoclasts at a density of 12,500 cells/well in Osteoclast Medium with 0.5% FBS supplemented with 25 ng/mL rhM-CSF. 

Osteoclast mono-cultures or co-cultures with osteoblasts were then stimulated with BC secretome and Osteoclast Medium with 0.5% FBS at a 1:1 ratio, supplemented with 25 ng/mL rhM-CSF. Secretome stimulation was performed only once, and the culture was maintained for 3 days. At the end of culturing, the conditioned medium of osteoclasts was collected and centrifuged at 4 °C at 400× *g* for 5 min and stored immediately at −80 °C.

For resorption quantification, bone slices were washed with deionized water and scrapped with cotton swabs to remove the cells from the bone surface. Osteoclast resorption events were stained with toluidine blue (Sigma-Aldrich, Cat#89640, Lyon, France). The entire bone surface was analyzed using a G50 100 point graticule (Pyser Optics, Cat#01A20.4075, Kent, UK) installed on the ocular of an BH-2 optical microscope (Olympus), as previously described [27]. Briefly, each intercept of a graticule point with a resorption event was counted as a hit, and the total number of hits throughout the bone surface were counted using the graticule as a frame (using a total of 16–17 graticules per bone slice). The total percentage eroded surface was estimated by dividing the total number of hits by the number of graticule grids used for the quantification. Individual resorption events were divided into two resorption types, namely pits and trenches. Pits are single, circular excavations with well-defined edges, while trenches are elongated and continuous grooves with a length/width ratio equal or greater than two [28]. The percentage of trenches per total event was used to compare individual experiments independently of eroded surface variations. Samples were blinded before eroded surface quantification.

### 2.5. TRAcP Activity Quantification

The activity of tartrate resistant acid phosphatase (TRAcP) present in the conditioned medium of osteoclast cultures was colorimetrically quantified. *p*-nitrophenyl phosphate (Merck, Cat#4876, Lyon, France) was added in the presence of 25 nM sodium tartrate (Merck, Cat#1066640100, Lyon, France) to the osteoclast conditioned medium, as previously described [29]. Absorbance was read at 405 nm using a Synergy Mx (BioTek, Porto Salvo, Portugal) plate reader. For normalization, total protein concentration was quantified using the DC Protein Assay (Bio-Rad, Cat#5000116, Lisbon, Portugal), according to the manufacturer’s instructions.

### 2.6. Quantitative Real-Time PCR (qRT-PCR) Analysis

MDA-231 and MDA-1833 cells were seeded in T75 flasks (Sarsted) and cultured as described above. When 80% confluence was reached, cells were washed with PBS and lysed with Trizol (Invitrogen, Cat#15596-018) and kept on ice. Total RNA from six independent replicates was extracted using the Direct-zol™ RNA miniPrep, according to the manufacturer’s protocol (Zymo Research, Tustin, CA, USA, Cat#ZY-R2052). RNA final concentration and purity (OD_260/280_) was determined using a NanoDrop 2000 instrument (NanoDrop Technologies, Wilmington, NC, USA). RNA was reverse transcribed into cDNA using the NZY First-Strand cDNA Synthesis Kit (NZYTech, Cat#MB12501, Lisbon, Portugal), according to the manufacturer’s protocol. qRT-PCR experiments were run using an iCycler iQ5 PCR thermal cycler (Bio-Rad Laboratories, Lisbon, Portugal) and analyzed with the iCycler IQ^TM^ software (Bio-Rad). A personalized PrimePCR array (Bio-Rad Laboratories, Lisbon, Portugal) was designed to analyze the range of adrenergic receptors. Target gene expression was quantified using the cycle threshold (Ct) values and relative mRNA expression levels were calculated as follows: 2^(Ct reference gene − Ct target gene). Human β2-microglobulin (β2M) and glyceraldehyde 3-phosphate dehydrogenase (GAPDH) were used as reference genes. Both target and reference genes were amplified with efficiencies between 100 ± 5%.

### 2.7. Immunocytochemistry

For β_2_-AR immunocytochemistry, MDA-231 and MDA-1833 cells were cultured in a multi-well μSlide (Ibidi) until semi-confluence, after which they were washed twice with PBS and fixed with 4% PFA for 10 min. After another wash with PBS, cells were permeabilized with cold 1% (*v*/*v*) Triton X-100 in PBS for 5 min. Unspecific binding was then blocked with 0.1% BSA solution in PBS for 1 h at 37 °C. Cells were incubated with a polyclonal rabbit anti-β_2_-AR (ProteinTech, Cat#13096-1-AP, RRID:AB_2225401, Manchester, United Kingdom) in a dilution of 1:100 in 0.1% BSA/PBS solution overnight at 4 °C. Excess antibody was washed with PBS and cells were then incubated with donkey anti-rabbit AlexaFluor 568 (ThermoFisher, 1:1000 dilution, Cat#A10042, RRID:AB_2534017, Carcavelos, Portugal) for 30 min at RT. After washing excess antibody with PBS, cells were counterstained with DAPI solution in PBS (1:1000 dilution) for 5 min. Samples were kept in PBS at 4 °C in the dark until image acquisition. Images were acquired in a SP2 confocal microscope (Leica, Vienna, Austria). Representative images were taken with a 40× objective at a resolution of 2048 × 2048 pixels. Brightness was adjusted with ImageJ software (v. 1.53t, NIH, Washington, DC, USA).

### 2.8. Western Blot

For protein collection, BC cells were cultured seeded at a density of 10^4^ cells/cm^2^ and left to proliferate until reaching ~80% confluence. Cells were then washed in cold PBS and lysed with cold RIPA buffer together with protease and phosphatase inhibitors (Sigma-Aldrich, 1:100 dilution, Cat#P8340 and Cat#P5726, Lyon, France) for 30 min on ice with on an orbital shaker. Lysate was collected and sonicated with three cycles of 3 s each. Samples were then centrifuged at 13,000× *g* for 10 min at 4 °C to pellet cell debris. Protein in the supernatant was then quantified using a DC Protein Assay Kit (Bio-Rad, Lisbon, Portugal). A total of 15 μg of protein from each cell lysate were prepared in non-reducing loading buffer and denatured for 5 min at 95 °C. Protein lysates were then loaded and resolved in pre-cast Bolt™ 10% polyacrylamide gels (Invitrogen, Cat#NW00105BOX, Carcavelos, Portugal). Separated proteins were then dry-transferred to nitrocellulose membranes using an iBlot 2 dry blotting system (ThermoFisher, Carcavelos, Portugal). Membranes were then blocked with 5% non-fat dry milk solution for 1 h at RT, followed by incubation in primary rabbit β_2_-AR antibody (ProteinTech, Cat#13096-1-AP, RRID:AB_2225401, Manchester, United Kingdom) at 1:1000 dilution in 5% milk solution overnight at 4 °C with gentle agitation. Membranes were incubated with HRP-conjugated anti-rabbit secondary antibody (Santa Cruz Biotechnology, 1:10,000 dilution, Cat#sc-2030, RRID:AB_631747, Heidelberg, Germany) for 1 h at RT under gentle agitation, followed by incubation in ECL substrate (GE Healthcare, Cat#1705061, Lyon, France) and chemiluminescence detection for 10 s with autoradiographic films (GE Healthcare, Lyon, France). Films were scanned on a GS-800 imaging densitometer (Bio-Rad, Lisbon, Portugal).

### 2.9. Proteomic Analysis

Conditioned medium from three replicate samples of MDA-231 or MDA-1833 treated with ISO as described in previous sections was collected and centrifuged at 300× *g* for 5 min to pellet cellular debris. Conditioned media were then transferred to microtubes and protein concentration was measured as described above. A total of 50 µg of protein from each condition was processed using the solid-phase-enhanced sample preparation (SP3) protocol, as previously described [30], followed by enzymatic digestion overnight with trypsin/LysC (2 micrograms) at 37 °C and 1000 rpm.

Protein identification was carried out by nano-liquid chromatography coupled with mass spectrometry (LC-MS/MS) and data were analyzed with Proteome Discoverer software (Thermo Scientific, v2.4), as described by Osório et al. [31]. Protein abundances were used to compare between conditions.

### 2.10. Statistics

All experiments with primary human cells were performed at least five times. Five replicates were used in resorption and TRAcP activity quantification, while for differentiation quantification three replicates were used, since six images were generated for each replicate. Experiments were randomized and data from resorption experiments were single-blinded (blinded by D.M.S. and quantified by F.C. and S.T.), while differentiation was quantified by F.C. and S.T. in an unsupervised fashion using the algorithm described above. qRT-PCR and proteomic screening experiments were performed with three independent experiments due to the small variability inherent in the use of cell lines. 

Non-parametric paired Friedman’s test followed by Dunn’s multiple comparison test were used to compare between conditions unless otherwise stated. Multiple comparisons were only performed if the test was statistically significant. Differences between groups were considered significant when * *p* < 0.05, ** *p* < 0.01 and *** *p* < 0.001. Data analysis was performed using GraphPad Prism software v.9.2.0. for Windows (GraphPad Software, RRID:SCR_002798). Comparison between clinical parameters of BC patients was performed using cross-tabulation analysis with the χ2-test for trend or Fisher’s exact test using GraphPad Prism v9.2.0. No data were excluded unless otherwise stated, and all outliers were included in the analysis.

In the proteomic screening, protein abundances were compared between conditions with Student’s *t*-test using Proteome Discoverer Software (Thermo Scientific, v2.4, RRID:SCR_014477). Heat maps and volcano plots comparing protein abundance between conditions were built with GraphPad v9.2.0. Only proteins that were present in every replicate of at least one condition were considered. Networks of biological processes associated with deregulated proteins between groups were constructed using the ClueGO plugin (v2.5.8, RRID:SCR_005748) from the Cytoscape software [32] (v3.9.0, RRID:SCR_003032) with the following parameters: GO Biological Processes ontology and “All Evidence” were selected; two-sided hypergeometric test and Bonferroni step down statistical options were used and only pathways with *p* < 0.05 were shown; minimum and maximum tree intervals were 3–6, with a minimum number of three genes; and 4% of genes selected for gene ontology terms, with a kappa score of 0.4. In proteomic data, *p*-values and abundance ratios in heat maps and volcano plots were normalized in order to be properly depicted (log_10_ for *p*-values and log_2_ for abundance ratios).

Principal component analysis (PCA) was performed with the Proteome Discoverer Software. Protein enrichment analysis was performed with WebGestalt toolkit (RRID:SCR_006786) [33]. Over-representation analysis (ORA) was used to analyze each set of deregulated proteins with the following conditions: the minimum and maximum number of genes was 5 and 2000, respectively; significance was set to *p* ≤ 0.05 with adjustments following the Benjamini–Hochberg method; gene ontology (biological process, cellular component, molecular function), pathway (Kyoto Encyclopedia of Genes and Genomes, Panther, Reactome and Wikipathway) and disease (Disgenet, GLAD4U and OMIM) functional databases were considered for the analysis.

## 3. Results

### 3.1. Primary and Advanced BC Patients Exhibit Increased Plasma Epinephrine Levels

We used the circulating epinephrine levels as a measure of the sympathetic tone in the plasma of primary (Stage I) and advanced (Stage IV) BC patients. We first characterized our study population according to several clinical parameters and found significant differences in tumor grade and age between primary and advanced BC patients, with no differences in menopausal status and tumor molecular subtype (Table 1). Subsequently, we proceeded to compare plasma epinephrine levels of BC patients and control individuals and found that both groups of BC patients displayed significantly increased levels of plasma epinephrine (Table 2). Interestingly, there is an exacerbation of circulating epinephrine release as the disease is established when compared to control individuals, but no significant differences were found between primary and advanced BC patients (*p* = 0.6347).

Since epinephrine binds to all ARs, the augmented levels of epinephrine in BC patients could affect BC cellular behavior. Thus, prior to in vitro functional studies, we proceeded to screen the expression of ARs in representative cell lines of parental and metastatic BC, MDA-231 and MDA-1833, respectively. MDA-1833 are variants of the MDA-231 cell line that metastasize specifically to bone and were previously characterized [34]. The gene and protein expressions of β_2_-AR in MDA-MB-231 and MDA-1833 cells showed that these cells predominantly express β_2_-AR, and cAMP accumulation assays demonstrated that these cells are responsive to sympathetic stimuli (Figure 1).

### 3.2. β-AR Signaling Induces a Shift in Parental MDA-231 Protein Secretion toward Osteoclastogenesis Inhibition

BC cells were previously reported to induce changes in bone metabolism at a distance while still in the primary tumor site [35]. Thus, we asked whether β_2_-AR signaling could modulate BC protein expression and affect bone metabolism. Using a proteomic approach, we screened the secretome of MDA-231 treated with the β-AR agonist ISO and compared its proteomic profile with the one from MDA-231 treated with vehicle control (Figure 2a). We detected a total of 64 proteins with significant deregulated expression (28 of them were upregulated and 36 downregulated in ISO-treated MDA-231 cells) (Figure 2b, Appendix A). Gene ontology network analysis as well as ORA showed that deregulated proteins were significantly associated with biological processes involved in extracellular matrix remodeling (Figure 2c,d).

When analyzing the set of proteins identified, we found that known osteoclast inhibitors Stanniocalcin-1 (STC-1) [36] and Clusterin (Clu) [37] were upregulated in ISO-treated MDA-231 cells. Conversely, osteoclastogenesis promotors cellular communication network 2 (CCN2) [38] and annexin II (ANXA2) [39] were downregulated with ISO treatment (Figure 2b). These results suggest that MDA-231 secretome under β-AR signaling might modulate osteoclastogenesis and could impact bone cell metabolism.

### 3.3. Secretome from Parental MDA-231 Cells under β_2_-AR Activation Impairs Osteoclast Differentiation and Resorption Activity

Since the expression of proteins known to be involved in bone metabolism was modulated after β-AR activation, we next proceeded to verify whether the changes in proteomic profile translated into functional significance.

#### 3.3.1. Osteoclast Monocultures

First, in order to ascertain the effect of MDA-231 secretome on osteoclast differentiation, human pre-osteoclasts were incubated with the secretome of MDA-231 cells. Human mature osteoclast number and area were then quantified (Figure 3a). Consistent with the proteomic screening results, the secretome of MDA-231 cells treated with ISO led to a 42% and 31% decrease in osteoclast numbers (cells with more than three nuclei) when compared to DMEM and unstimulated controls, respectively (Figure 3b,c). β2-AR signaling was directly implicated in the observed effect since osteoclast numbers were no different from controls when ICI, a specific β2-AR antagonist, was added to MDA-231 together with ISO (Figure 3b,c). Coherent with the osteoclast differentiation quantification, osteoclast specific cathepsin K gene expression followed the same profile after stimulation with ISO-treated MDA-231 secretome (Appendix A). On the other hand, when comparing the area of osteoclasts in the different conditions, no differences were observed (Figure 3d).

Since BC secretome could also directly affect the resorption activity of osteoclasts after they are matured and fully functional, we proceeded to incubate mature osteoclasts with MDA-231 secretome and quantified the extent of bone resorption in the absence of RANKL (Figure 3e). Similarly to the osteoclastogenesis assays, we observed a significant decrease in the resorption activity of 61%, when osteoclasts were incubated with the secretome from MDA-231 cells treated with ISO (Figure 3f,g). This effect was mediated by β2-AR, since the addition of ICI abrogated this effect. Osteoclasts are able to resorb the bone surface through two characteristic modes of resorption: they resorb the bone while being static or while moving through the surface of the bone, generating resorption pits or trenches, respectively [40]. Resorption trench mode is inherently faster and favors bone fragility [24], and therefore, differences in trench content could reflect changes in osteoclast aggressiveness. No significant differences were found in trench resorption content when comparing between conditions (Figure 3h). The observed decrease in total resorption was partially due to a 54% reduction in TRAcP activity, an osteoclast-specific enzyme involved in collagenolysis (Figure 3i).

#### 3.3.2. Osteoblast–Osteoclast Co-Cultures

In the bone microenvironment, osteoclasts receive input from neighboring osteoblasts that promote osteoclast differentiation and activation. Aiming to replicate this interaction, a co-culture of mature human osteoclasts and human osteoblasts was established. Osteoblasts were positive for the osteoblast marker alkaline phosphatase (ALP) and several calcium deposits were observed by alizarin red staining after 14 days of culture (Appendix A). We confirmed that these cells expressed RANKL at their cellular membrane and were able to increase osteoclast resorption activity in several independent experiments (Appendix A). Osteoclast and osteoblast co-cultures were exposed to the secretome from ISO-treated MDA-231 cells and bone resorption activity was evaluated (Figure 4a). Contrarily to what was observed in osteoclast monocultures, no statistically significant differences were observed between conditions in total resorption activity (Figure 4b,c), trench percentage (Figure 4d) or TRAcP activity (Figure 4e). Therefore, our results suggest that osteoblast-derived factors in the co-culture model rescued the resorption activity of osteoclasts.

### 3.4. Secretome of Bone Tropic MDA-1833 under β2-AR Activation Does Not Affect Human Osteoclast Differentiation and Resorption Activity

Bone is a highly innervated tissue, and sympathetic neurons are found across the periosteum and around blood vessels in the bone marrow [41]. In addition, bone marrow-resident cells were reported to express tyrosine hydroxylase (TH) and synthesize catecholamines [42,43]. Therefore, in addition to the increased epinephrine levels in the plasma of metastatic BC patients, the local production of catecholamines might also contribute to the development of metastatic foci during BC bone metastasis. It is then of crucial importance to understand how the adrenergic signaling modulates the bone metastatic niche.

#### 3.4.1. Osteoclast Monocultures

During the course of extravasation and metastasis, BC cells acquire a set of characteristics that differ from the primary tumor. In order to ascertain whether bone tropic cells elicit different outcomes in osteoclast differentiation than its parental counterpart, pre-osteoclasts were exposed to β_2_-AR primed MDA-1833 secretome in a similar experimental layout, as previously carried out with MDA-231 cells (Figure 5a). Contrary to the effect of MDA-231 cells, secretome from ISO-treated MDA-1833 cells did not affect the number of osteoclasts nor the osteoclast area (Figure 5b–d).

Following the same reasoning as that of the MDA-231 secretome experiments, we proceeded to quantify mature osteoclast resorption activity under the effects of MDA-1833 secretome instead (Figure 5e). No differences were observed in the percentage of eroded surface when comparing the effects of ISO-treated MDA-1833 secretome with their respective controls (Figure 5f,g). However, ISO-treated MDA-1833 secretome significantly blunted the ability of osteoclasts to resorb in trench mode in a β_2_-AR-dependent fashion (36% reduction, Figure 5h). Nonetheless, this was not due to changes in osteoclastic TRAcP activity, since no significant differences were found between conditions (Figure 5i).

#### 3.4.2. Osteoblast–Osteoclast Co-Cultures

Our findings show that the secretome of MDA-1833 under β_2_-AR activation did not significantly impact human osteoclast differentiation, although we were able to observe changes in trench resorption events. We then asked whether RANKL-producing osteoblasts would change the dynamic of the interaction between MDA-1833 conditioned medium and osteoclasts (Figure 6a). Similarly to osteoclast monocultures, conditioned medium from ISO-treated MDA-1833 did not induce changes in osteoclast resorption activity in a osteoblast/osteoclast co-culture setting (Figure 6b,c). However, osteoblasts rescued the normal trench-forming ability of osteoclasts, since no changes in trench percentage were observed when ISO-treated MDA-1833 secretome was used in a co-culture setting (Figure 6d). Furthermore, similarly to osteoclast monocultures, no significant differences in TRAcP activity were observed at the end of the experiment (Figure 6e).

In conclusion, secretome from MDA-1833 cells under β_2_-AR activity led to a decreased osteoclast aggressiveness, as evidenced by a reduction in trench resorption percentage, but not in terms of the total area of resorption events. This was only seen in osteoclast monocultures, since in the presence of osteoblasts no differences were observed. Furthermore, MDA-1833 secretome did not affect osteoclast differentiation.

### 3.5. Bone Tropic MDA-1833 and Its Parental Counterpart MDA-231 Express Different Levels of Osteoclastogenic Factors under β_2_-AR Activation

Functional studies on osteoclast activity showed that, contrary to the parental MDA-231 cells, bone tropic MDA-1833 cells do not impair osteoclast differentiation and activity. In order to understand what was driving these different outcomes, we first asked what the main differences between both cell lines regarding protein expression under β_2_-AR activation were. The proteomic screening of ISO-treated MDA-1833 secretome was performed and compared to the secretome from ISO-treated MDA-231 cell lines (Figure 7a,b). A total of 69 proteins were significantly deregulated between ISO-treated MDA-1833 and MDA-231 cell lines (31 upregulated and 38 downregulated proteins in ISO-treated MDA-1833) (Figure 7c, Appendix A). Of note, the proteins identified in proteomic analysis of ISO-treated MDA-231 cells STC-1 and CCN2 were also differentially expressed between ISO-treated MDA-1833 and ISO-treated MDA-231 secretomes. In addition, insulin-like growth factor binding protein 7 (IGFBP7) and cystatin C (CST3), proteins previously described as inhibiting osteoclastogenesis [44,45], were downregulated in MDA-1833 samples. These results are consistent with the previous functional assays and emphasize the differential responses to β_2_-AR signaling in the parental cell line MDA-231 and the bone tropic cell line MDA-1833. Furthermore, proteins that were differentially expressed in parental and bone tropic BC cell lines under β_2_-AR signaling were associated with biological processes such as platelet degranulation, extracellular structure organization and immune modulation (Figure 7d,e).

## 4. Discussion

The metastatic dissemination of BC cells from the primary tumor to distant sites is the main cause of death of BC patients, with the bone being the most common site of metastasis. Intracardiac injection models as well as xenograft denervation studies in vivo have shown that β_2_-AR signaling exerts a pro-metastatic, bone colonization and immunomodulatory role in BC [14,15]. However, the interactions between BC cells and bone cells in a human context are still poorly understood. In this study, using a combination of clinical data, proteomic screening and in vitro functional assays with human primary cells, we investigated the effect of sympathetic signaling in the crosstalk taking place between either parental or metastatic BC cells and bone cells (Figure 8).

The increased epinephrine content during BC establishment and progression that we observed highlight the need for a better understanding of the role of the SNS activity in BC progression and bone metastasis. In particular, β_2_-AR signaling was reported to be the major contributor for BC metastasis in in vivo models of chronic stress and SNS hyperactivity [14,46,47]. Therefore, we focused on high β_2_-AR-expressing parental MDA-231 and bone tropic MDA-1833 cells to further explore the β_2_-AR signaling pathway in a bone metastatic context. 

We showed that the secretome of MDA-231 cells exhibits significant differences in terms of protein expression after the activation of β_2_-AR signaling. In particular, STC-1 and Clu were upregulated in the MDA-231 secretome under sympathetic signaling. STC-1 is a known osteoclast inhibitor, since transgenic mice expressing STC-1 under a muscle-specific promotor display increased cortical and trabecular bone thickness concomitant with a decreased osteoclast activity [36]. Furthermore, STC-1 was shown to be directly involved in murine osteoclastogenesis suppression in vitro [48]. Similarly, secreted Clu was shown to hamper osteoclastogenesis in murine osteoclast precursors [37].

The upregulation of osteoclast inhibitors in MDA-231 secretome was consistent with the observed decrease in human osteoclast differentiation and bone matrix resorption activity in our model, in up to six independent osteoclast donors with blinded or unsupervised analysis. However, the crosstalk of osteoclasts with human osteoblasts mitigated this effect. In contrast, paracrine signaling from metastatic MDA-1833 had no effect either on osteoclast monocultures or in co-culture with osteoblasts. Given that osteoclast differentiation and activity is tightly regulated by osteoblasts in the bone niche, our results suggest that the sympathetic modulation of BC paracrine signaling does not affect osteoclast bone resorption activity and we would expect no differences in bone turnover in BC patients. Our findings are in agreement with a recent study by Chiou et al., where the authors demonstrate that the injection of MDA-MB-231 secretome in immunocompromised mice led to increased bone formation in the pre-metastatic bone niche, with no changes in osteoclast activity [49]. This altered bone matrix deposition could facilitate BC extravasation into the bone niche prior to metastatic foci establishment [49]. However, we did not recapitulate the observations from previous studies that show the modulation of bone resorption activity by BC cells, even before metastatic spread has occurred [35,50]. This might be an acceptable outcome in primary BC patients with no major alteration in bone turnover; however, it is paradoxical in stage IV metastatic BC patients who exhibit metastatic bone foci and extensive bone lesions. Moreover, previous studies indicate an exacerbation in bone destruction after β_2_-AR agonism, in the immunosuppressed mice models of breast cancer bone metastasis as well as mice in in vivo and in vitro models of the pharmacological activation of β_2_-AR [14,51,52,53].

Despite the fact that our findings are not aligned with these studies, species related differences in bone cell metabolism could explain the observed differences. It is widely accepted that BC secretome increase osteoclast differentiation and resorption in experiments using cells from mice [50,54,55,56,57]. In contrast, the results obtained from experiments with human cells are not so homogeneous, with some studies reporting increases in osteoclastogenesis [54,58] while others show decreased resorption activity [59]. Additional evidence of distinct outcomes in human and mouse cells arise from the observation that there is a significant increase in RANKL promotor activation in murine bone marrow stromal cells after exposure to paracrine signaling from BC cells (Appendix A). When assessing osteoclast activity in our model of human osteoblast/osteoclast co-cultures, no differences were observed in terms of matrix degradation after incubation with BC secretome. Furthermore, although sympathetic activation was reported to directly increase osteoclastogenesis in murine in vitro models [60,61], no alterations in differentiation and resorption activity were apparent after the daily activation of β_2_-AR in human osteoclasts/osteoblasts (Appendix A). Again, direct β_2_-AR stimulation in murine bone marrow stromal cells translated into increased RANKL promotor activation, while no differences were observed in human osteoblast RANKL production (Appendix A). To our knowledge, we are the first to describe the effect of BC paracrine signaling on human osteoclast activity under β_2_-AR agonism in vitro, and therefore, species-related differences should be accounted for when comparing with in vivo studies. 

The bone metastatic niche is composed of multiple cell types that can contribute to the establishment of osteolytic bone metastasis [4]. We observed the consistent and robust decrease in osteoclast differentiation and activity after incubation with conditioned medium from MDA-231 cells under β_2_-AR signaling. The inclusion of other cellular players, such as cancer-associated fibroblasts [62], endothelial cells [63] and a functional immune system [64], could change the dynamic of SNS activation in the bone metastatic niche. In fact, osteocytes account for the majority of osteoblast-lineage cells in bone, and osteocyte-like cell lines have already been described to increase osteolytic output in response to SNS activation [65]. Future studies should be conducted to analyze the contribution of other cell types in the dynamics of the SNS modulation of BC bone metastasis and the exacerbation of the metastatic vicious cycle of bone degradation.

Proteomic screening identified multiple differentially expressed proteins between MDA-231 and MDA-1833 cells, similarly to what is described in other previous proteomic analysis studies [66,67]. In addition, the β2-AR activation of osteoclast STC-1 upregulation and CCN2 downregulation were already described in parental MDA-231 cell lines stimulated with ISO [68]. These observations suggest that, under high sympathetic activity, the capacity of BC cell lines to modulate osteoclast activity is altered when they acquire metastatic traits, since bone tropic MDA-1833 paracrine signaling does not elicit osteoclast bone resorption impairment in our model. Similar adaptations are found in brain metastatic cell lines, where BC cells gain the ability to interact with their surrounding tissue, which is inherently different from the primary tumor site [69,70]. Together with data from other studies that show the increased migratory capacity of metastatic BC cell lines after β_2_-AR agonism [71], our observations point to distinct β_2_-AR downstream effectors in bone tropic cells that could putatively facilitate the colonization of the bone microenvironment under high sympathetic input when compared to its parental cell line.

BC bone metastases lead to extensive bone degradation and several in vivo studies suggest that the bone is already conditioned by BC at earlier stages of the disease [35,49]. Interestingly, there is an apparent mismatch between the distinct modulation of osteoclast activity by β_2_-AR primed parental or metastatic BC cells and the lack of differences in circulating epinephrine content observed in primary and advanced BC patients. This could be reflective of the following: the absence of stimuli from neighboring immune cells, fibroblasts or hematopoietic/mesenchymal stem cell niches in our in vitro model; the distinct targets of BC secretome after sympathetic signaling in cells other than osteoclasts; or the intrinsic decrease in in vivo primary BC cell β_2_-AR expression that would explain the conditioning of the bone microenvironment in earlier stages of BC. In fact, a discrepancy between AR expression profile in common BC cell lines and human tumors was previously identified [72], and strong tumor β_2_-AR expression was correlated with increased disease-free survival [72,73]. β_2_-AR was also shown to be downregulated in BC tumors when compared to normal breast tissue in a BC gene expression database [74], suggesting that β_2_-AR signaling is decreased as the disease progresses. 

Some limitations apply to our study. In our clinical samples, age-specific differences between groups cannot be excluded, since primary BC patients were significantly older than advanced BC patients and control individuals. Furthermore, access to patient medication and co-morbidities at the time of diagnosis and blood collection was limited, and obesity and diabetes, as well as the intake of antihypertensive, analgesic or anxiolytic drugs, could affect catecholamine synthesis or release into the circulation [75,76,77,78,79]. These limitations should be addressed in a larger, randomized clinical trial with patients matched for co-morbidities and treatment regimens. In addition, further in vitro and in vivo mechanistic validation studies are required to determine the potential clinical benefit of targeting the proteins identified in this study. Finally, other ARs besides β_2_-AR might be involved in the BC response to epinephrine in vivo, which prompts the investigation of adrenergic signaling in other BC cell lines, such as MCF7, BT-474 or MDA-MB-453, and their bone tropic counterparts in future studies.

## 5. Conclusions

In summary, we present new data concerning the changes in circulating epinephrine that take place during BC and highlight the effect of β2-AR signaling on the crosstalk between BC cells and bone niche cells, osteoclasts and osteoblasts. Interestingly, although epinephrine levels in BC patients are maintained throughout disease progression, proteomic screening and functional studies identified the distinct outcomes of β2-AR activation after BC metastasis. We showed that, contrary to what is expected, parental BC cells decrease osteoclast differentiation and resorption activity in vitro under β2-AR signaling activation. In contrast, the secreted factors from β2-AR primed metastatic BC cells do not affect osteoclast differentiation and resorption in our model, suggesting that the secreted factors from BC cells after β2-AR agonism are not directly involved in the local exacerbation of bone resorption activity. Future studies should focus on the study of cancer/osteoblast/osteoclast direct cell–cell interactions and the role of other cell types present in the bone marrow in the dynamics of the SNS activation of the bone metastatic niche.

## Figures and Tables

**Figure 1 biomolecules-13-00622-f001:**
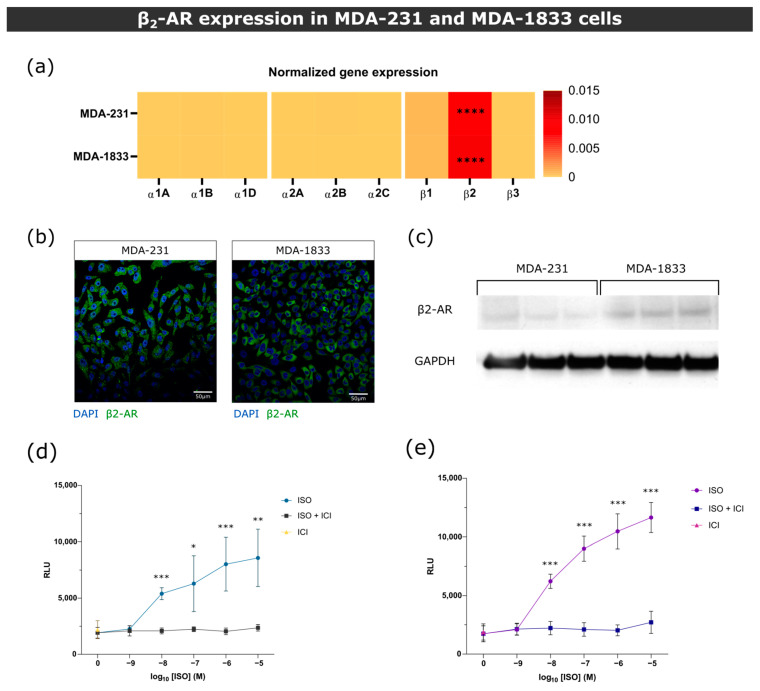
β2-AR gene and protein expression in MDA-231 and MDA-1833 cells. (**a**) Normalized gene expression of ARs in primary MDA-231 and metastatic MDA-1833 cell lines performed by real time PCR. One-way ANOVA with Holm–Šídák’s multiple comparison test was used to compare β2-AR expression to every other ARs in MDA-231 or MDA-1833 (*n* = 6, **** *p* < 0.0001). No significant differences were found in AR expression between MDA-231 and MDA-1833 cells (two-way ANOVA with Šídák’s multiple comparison test). (**b**) Representative micrograph of the expression of β2-AR on MDA-231 cells (left) and MDA-1833 cells (right) by immunocytochemistry. Nuclei—blue; β2-AR—green. Scale bar—50 μM. (**c**) Expression of β2-AR on MDA-231 and MDA-1833 cells by Western blot. Uncropped blot can be found in Appendix A. *(***d**) Stimulation of MDA-231 or (**e**) MDA-1833 cells with ISO results in cAMP accumulation, which is abrogated by the incubation with 1 μM ICI. Data were generated from three experiments performed in triplicate (Mann–Whitney test, * *p* < 0.05, ** *p* < 0.01, *** *p* < 0.001).

**Figure 2 biomolecules-13-00622-f002:**
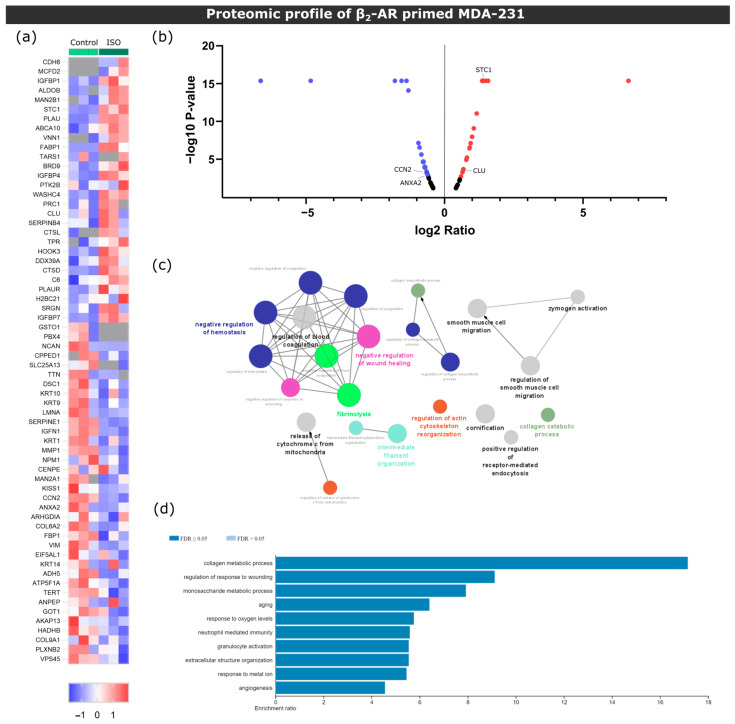
Proteomic screening of MDA-231 conditioned media under β2-AR activation. (**a**) Heatmap depiction of significantly deregulated proteins found in conditioned medium from control or ISO-treated MDA-231 cells (Student’s *t*-test, *p* ≤ 0.05). The ratio of protein abundances was color coded as shown in the legend. Data from three independent samples are shown, one sample per column. Gray cells represent proteins not detected in a particular sample. (**b**) Volcano plot distribution of significantly deregulated proteins. The ratio between protein abundances between MDA-231 cells treated with ISO and control samples are plotted on the X axis, and the *p*-value is plotted on the Y axis. Red dots represent proteins with a fold change ≥ 1.5, while purple dots represent proteins with a fold change ≤ −1.5. (**c**) Gene ontology biological process network analysis of differentially expressed proteins in ISO-treated MDA-231 cells. Only networks with *p* ≤ 0.05 are shown (two-sided hypergeometric test and Bonferroni step-down correction). Different colors illustrate different biological processes, while the size of the nodes is proportional to the statistical significance of each process. (**d**) Significantly enriched biological process terms in the group of deregulated proteins in ISO-treated MDA-231 cells. ORA was used and only processes with a false discovery rate of ≤0.05 are shown.

**Figure 3 biomolecules-13-00622-f003:**
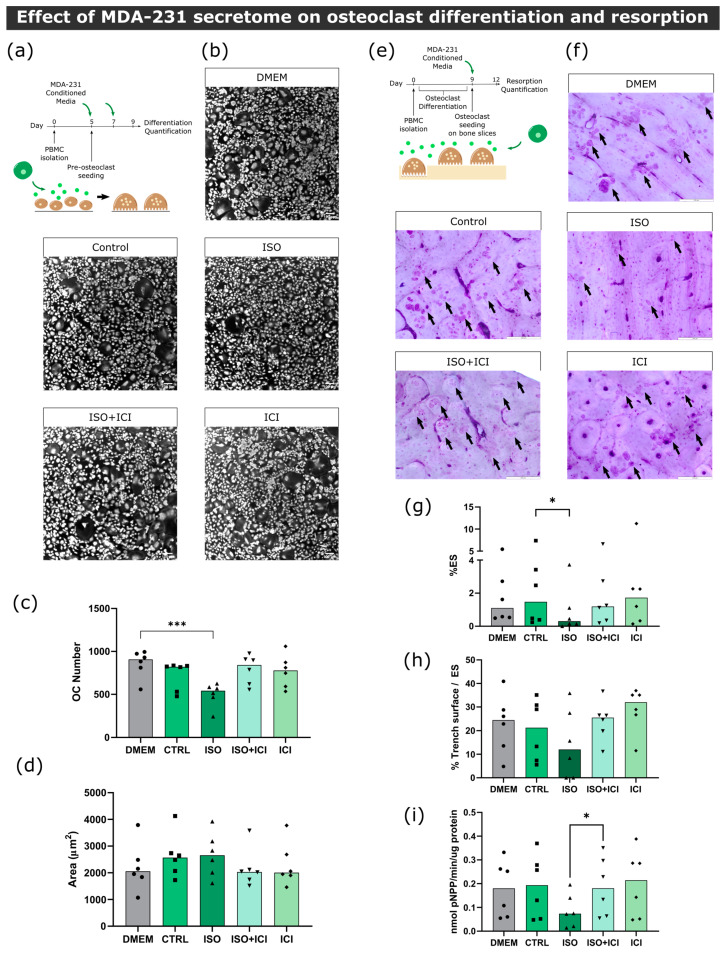
Effect of MDA-231 conditioned medium on osteoclast differentiation and resorption activity. (**a**) Timeline of the osteoclast differentiation assay. Pre-osteoclasts are seeded after five days of culturing and stimulated with MDA-231 cell-conditioned media for four more days, changing the media at day 7. (**b**) Representative micrographs of differentiated osteoclasts were stained with HCS CellMask Deep Red Stain. Scale bar—100 μM. (**c**) Osteoclast number and (**d**) area after four days of MDA-231-conditioned media stimulation. Osteoclasts with three or more nuclei were included in the analysis. Data are expressed as the median of individual data points from six independent experiments (Friedman’s test followed by Dunn’s multiple comparison test, *** *p* ≤ 0.001). (**e**) Timeline of the resorption assay. Osteoclasts were differentiated for nine days and seeded on top of bone slices, followed by incubation with MDA-231 cell-conditioned medium for another three days. (**f**) Representative micrographs of the surface of bone slices after three days of resorption. Resorption events are highlighted by black arrows. Scale bar—200 μM. (**g**) Estimated percentage of eroded surface, (**h**) trench percentage per total eroded surface and (**i**) TRAcP activity were quantified at the end of the experiment. Data was expressed as a median of individual data points from six independent experiments (Friedman’s test followed by Dunn’s multiple comparison test, * *p* ≤ 0.05).

**Figure 4 biomolecules-13-00622-f004:**
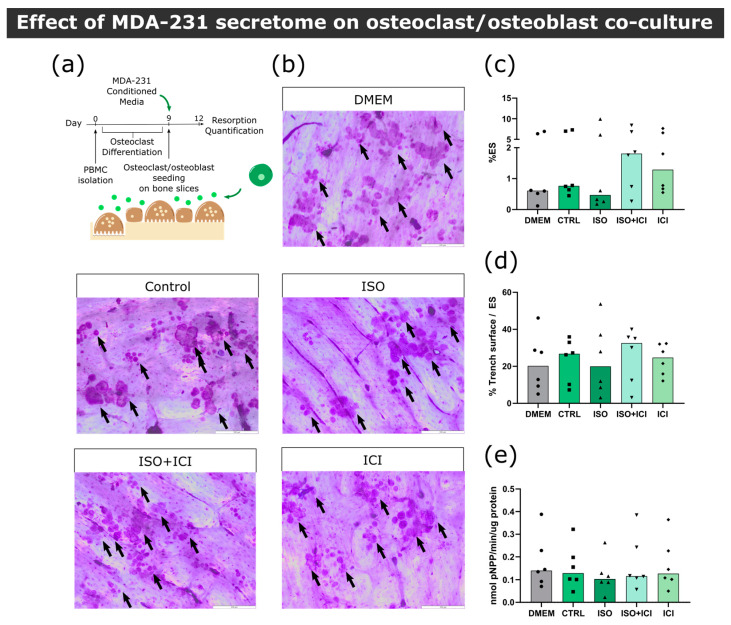
Effect of MDA-231 secretome on osteoclast/osteoblast co-culture. (**a**) Timeline of the resorption assay in a co-culture setting. Mature osteoclasts were seeded with osteoblasts and exposed to MDA-231 conditioned media for three days. (**b**) Representative micrographs of the surface of bone slices after three days of resorption in co-cultures. Resorption events are highlighted by black arrows. Scale bar—200 μM. (**c**) Estimated percentage of eroded surface, (**d**) trench percentage per total eroded surface and (**e**) TRAcP activity quantified at the end of the co-culture experiment. Data are expressed as the median of individual data points from six independent experiments (Friedman’s test followed by Dunn’s multiple comparison test, * *p* ≤ 0.05).

**Figure 5 biomolecules-13-00622-f005:**
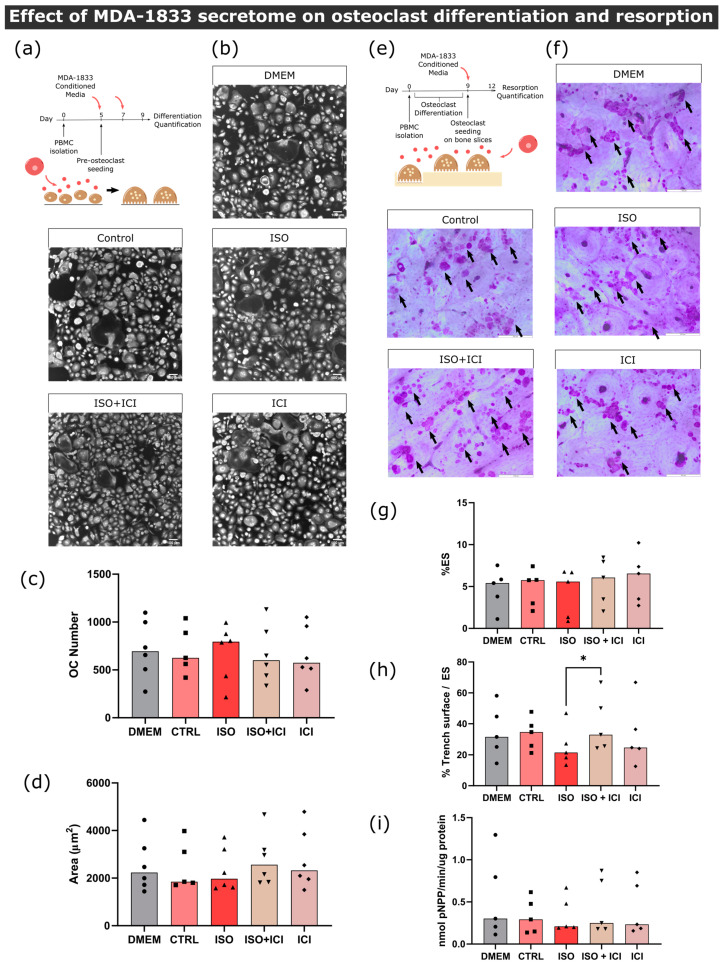
Effect of MDA-1833-conditioned medium on osteoclast differentiation and resorption activity. (**a**) Timeline of the osteoclast differentiation assay. Pre-osteoclasts are seeded after five days of culturing and stimulated with the MDA-1833 cell secretome for four more days, refreshing the media at day 7. (**b**) Representative micrographs of differentiated osteoclasts stained with HCS CellMask Deep Red Stain. Scale bar—100 μM. (**c**) Osteoclast number and (**d**) area after four days of MDA-1833 secretome stimulation. Osteoclasts with three or more nuclei were included in the analysis. Data are expressed as the median of individual data points from six independent experiments (Kruskal–Wallis test). One data point from the CTRL condition was excluded due to a technical error. (**e**) Timeline of the resorption assay. Osteoclasts are differentiated for nine days and seeded on top of bone slices, followed by incubation with MDA-1833 cell secretome for another three days. (**f**) Representative micrographs of the surface of bone slices after three days of resorption. Resorption events are highlighted by black arrows. Scale bar—200 μM. (**g**) Estimated percentage of eroded surface, (**h**) trench percentage per total eroded surface and (**i**) TRAcP activity quantified at the end of the experiment. Data are expressed as the median of individual data points from five independent experiments (Friedman’s test followed by Dunn’s multiple comparison test, * *p* ≤ 0.05).

**Figure 6 biomolecules-13-00622-f006:**
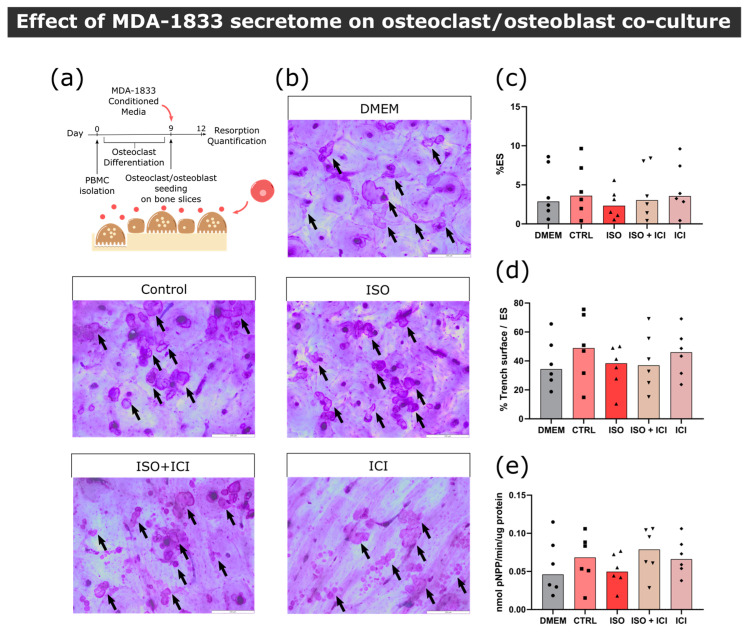
Effect of MDA-1833 secretome on osteoclast/osteoblast co-cultures. (**a**) Timeline of the resorption assay in a co-culture setting. Mature osteoclasts were seeded with osteoblasts and exposed to MDA-1833-conditioned media for three days. (**b**) Representative micrographs of the surface of bone slices after three days of resorption in co-cultures. Resorption events are highlighted by black arrows. Scale bar—200 μM. (**c**) Estimated percentage of eroded surface, (**d**) trench percentage per total eroded surface and (**e**) TRAcP activity quantified at the end of the co-culture experiment. Data are expressed as the median of individual data points from six independent experiments (Friedman’s test).

**Figure 7 biomolecules-13-00622-f007:**
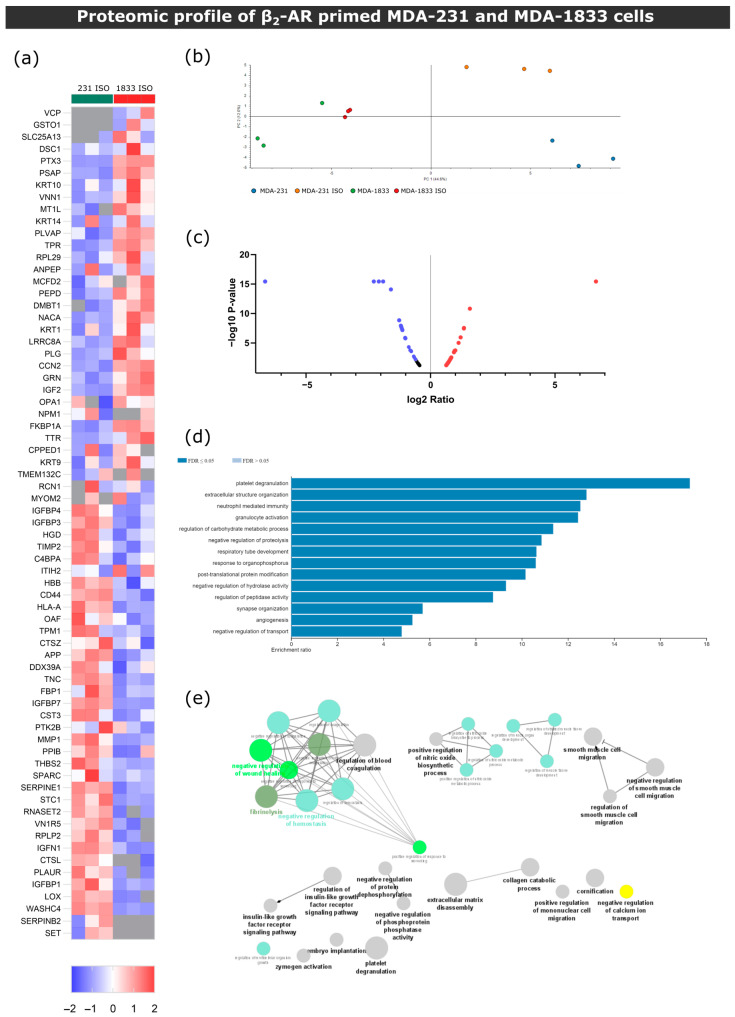
β2-AR signaling leads to different outcomes in parental MDA-231 or bone tropic MDA-1833 cells. (**a**) Heatmap depiction of significantly deregulated proteins found in conditioned medium from control or ISO-treated MDA-231 and MDA-1833 cells (Student’s *t*-test, *p* ≤ 0.05). The ratio of protein abundances was color coded as shown in the legend. Data from three independent samples are shown, one sample per column. Gray cells represent proteins not detected in a particular sample. (**b**) PCA of deregulated proteins between MDA-231 and MDA-1833, both in the control and ISO-treated conditions. (**c**) Volcano plot distribution of significantly deregulated proteins in ISO-treated MDA-1833 and ISO-treated MDA-231 samples. The ratio between protein abundances between ISO MDA-1833 cells and ISO MDA-231 cells is plotted on the X axis, and the *p*-value is plotted on the Y axis. Red dots represent proteins with a fold change ≤ 1.5, while purple dots represent proteins represent proteins with a fold change ≤ −1.5. (**d**) Significantly enriched biological process terms in the group of deregulated proteins in ISO-treated MDA-1833 and ISO-treated MDA-231 cells. ORA was used and only processes with a false discovery rate of ≤ 0.05 are shown. (**e**) Gene ontology biological process network analysis of differentially expressed proteins in ISO-treated MDA-1833 and ISO-treated MDA-231 cells is shown. Only networks with *p* ≤ 0.05 are shown (two-sided hypergeometric test and Bonferroni step-down correction).

**Figure 8 biomolecules-13-00622-f008:**
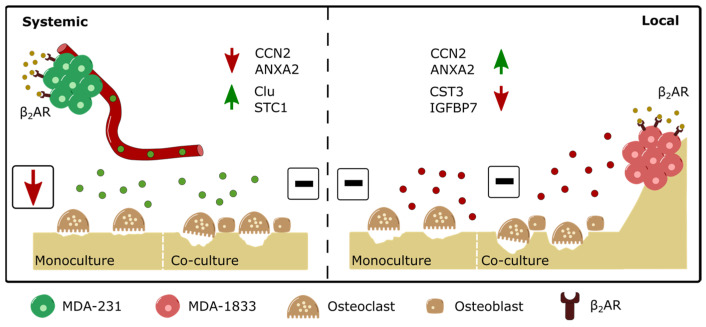
Schematic depiction of the differential modulation of osteoclast resorption activity by parental or bone tropic breast cancer cells. (Left panel) β2-AR stimulation drives the upregulation of several proteins, such as Clu and STC1, and the downregulation of proteins such as CCN2 and ANXA2 in parental MDA-231 BC cells. Although exposure to the secretome of β2-AR primed parental BC cells leads to decreased osteoclast resorption activity, the addition of osteoblasts to the model rescues osteoclast activity to control levels. (Right panel) on the other hand, the β2-AR stimulation of bone tropic MDA-1833 cells leads to the upregulation of proteins such as CCN2 and ANXA2 and the downregulation of proteins such as CST3 and IGFBP7. The distinct modulation of protein expression by β2-AR on bone tropic cells did not result in observable changes in osteoclast activity, either in monocultures or in co-cultures with osteoblasts.

**Table 1 biomolecules-13-00622-t001:** The clinical parameters of control individuals and BC patients included in our study.

Variable	Control Individuals (*n* = 12)	Stage I BC Patients (*n* = 12)	Stage IV BC Patients (*n* = 12)	*p* Value
Age				
<65	11	5	11	0.0094
≥65	1	7	1	
Primary tumor grade				
T1		12		
T2			1	
T3			4	<0.0001
T4			2	
N/A			5	
Menopause				
Pre-Menopause		1	5	0.3168
Post-Menopause		11	7	
Molecular Subtype				
Luminal A		4	4	
Luminal B		8	8	1.0000
Metastasis				
Bone only		N/D	7	
Bone and other sites		N/D	5	

N/D—not detected.

**Table 2 biomolecules-13-00622-t002:** Epinephrine plasma levels of control individuals and BC patients at different stages of progression.

	Plasma Epinephrine Concentration (pg/mL)	Adjusted *p*-Value
Control	22.04 ± 31.95	-
Stage I BC patients	105.9 ± 186.03	0.0329 *
Stage IV BC patients	143.8 ± 413.69	0.0091 **

Epinephrine concentration is expressed as mean ± standard deviation. Patient groups were compared with control using a Kruskal–Wallis test with Dunn’s multiple comparison test: * *p* < 0.05, ** *p* < 0.01.

## Data Availability

The data that supports the findings of these studies are available from the corresponding author upon reasonable request. The mass spectrometry proteomics data are openly available in the ProteomeXchange Consortium via the PRIDE [80] partner repository with the dataset identifier PXD030371 and 10.6019/PXD030371.

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
