# Peer review of "The Secretome of Parental and Bone Metastatic Breast Cancer Elicits Distinct Effects in Human Osteoclast Activity after Activation of β2 Adrenergic Signaling"

_biomolecules, 2023, doi:10.3390/biom13040622_

Round 1

Reviewer 1 Report

While a couple of figures are not convincing, overall the experiments/analysis performed is thorough and discussion is thorough. The paper provides important insight in the clinical significance and mechanism of b2AR in breast cancer bone metastasis.

Line 348 – MDA-MB-231 cell line is not from a primary tumor, it is from a pleural effusion (PMID: 4412247).  It is a metastatic cell line model (PMID: 23118918). The important distinction between the two metastatic cell line models – 231 and 1833 – is 1833 is a 231 subline that specifically honed to the bone, as opposed to other organs, such as lung.

https://www.mskcc.org/research-advantage/support/technology/tangible-material/mda-231-bom-1833-human-breast-adenocarcinoma-cell-line: “Description:MDA-231-BoM-1833 is a breast adenocarcinoma cell line established from bone metastasis of nude mice inoculated with the parent cell line. It is a triple negative breast cancer (TNBC) cell line.  This cell line is part of a panel of human breast cancer cell lines selected to metastasize to specific organs. This cell line was established from a 51-year-old female of Caucasian ethnicity and was derived from a metastatic site of pleural effusion.”

Figure 1A: qPCR data should be shown for all of the genes as bar graphs with error bars and p-values of multiple biological replicates (e.g., n=3). It the authors decide to keep the heatmap, the figure legend needs to indicate the number of biological replicates the heat map was generated from along with p-values comparing expression of 231 to 1883 and b2AR to the other receptors.

Figure 1B: Are the two images taken at the same exposure?  The 231 image looks like overexposed, nonspecific background staining.  The 1833 staining looks real. However, b2AR is a GPRC plasma membrane protein.  Authors should try immunostaining without permeablizing the cells, otherwise, they need to explain why the receptor is cytoplasmic in their cells.

Figure 1C: western blot is not convincing – bands could be simply background signal. Authors should consider running a control cell line that is negative to b2AR and positive for bAR alongside 231 and 1833. 

Figure 1D & E: statistical significance needs to be shown (e.g., pvalues comparing ISO and ISO + ICI)

Authors should consider including a model of what they are proposing the role of b2AR to be in bone honing and establishment of the metastatic niche. 

Reviewer 2 Report

See attached file

Round 2

Reviewer 2 Report

The authors have adequately addressed the comments raised to the original submission and toned down the clinical significance of their findings. The manuscript is improved as a result.